# A High-Throughput Assay for In Vitro Determination of Release Factor-Dependent Peptide Release from a Pretermination Complex by Fluorescence Anisotropy—Application to Nonsense Suppressor Screening and Mechanistic Studies

**DOI:** 10.3390/biom13020242

**Published:** 2023-01-27

**Authors:** Mikel D. Ghelfi, Saleem Y. Bhat, Hong Li, Barry S. Cooperman

**Affiliations:** Department of Chemistry, University of Pennsylvania, Philadelphia, PA 19104, USA

**Keywords:** premature termination codon, readthrough, termination, translation readthrough-inducing drug (TRID), high-throughput screening (HTS), ataluren

## Abstract

Premature termination codons (PTCs) account for ~12% of all human disease mutations. Translation readthrough-inducing drugs (TRIDs) are prominent among the several therapeutic approaches being used to overcome PTCs. Ataluren is the only TRID that has been approved for treating patients suffering from a PTC disease, Duchenne muscular dystrophy, but it gives variable readthrough results in cells isolated from patients suffering from other PTC diseases. We recently elucidated ataluren’s mechanism of action as a competitive inhibitor of release factor complex (RFC) catalysis of premature termination and identified ataluren’s binding sites on the ribosome responsible for such an inhibition. These results suggest the possibility of discovering new TRIDs, which would retain ataluren’s low toxicity while displaying greater potency and generality in stimulating readthrough via the inhibition of termination. Here we present a detailed description of a new in vitro plate reader assay that we are using both to screen small compound libraries for the inhibition of RFC-dependent peptide release and to better understand the influence of termination codon identity and sequence context on RFC activity.

## 1. Introduction

Premature termination codons (PTCs) arise in mRNA as a consequence of nonsense mutations. Such mutations lead to the replacement of an amino acid codon in mRNA by one of three stop codons, UAA, UGA or UAG [1,2,3,4]. The binding of the release factor complex (eRF1.eRF3.GTP, RFC) to a PTC results in the hydrolysis of a peptidyl-tRNA bound in the neighboring ribosome P-site and the release of a typically inactive truncated protein product, a process also known as termination. Nonsense mutations constitute ~20% of the inherited or de novo germline mutations [5,6,7]. Globally, there are ~7000 known genetically transmitted disorders in humans, and ~12% of all human disease mutations are nonsense mutations [8]. Clearly, millions of people worldwide would benefit from effective therapies directed toward PTC suppression. Many therapeutic approaches are being used to overcome PTCs [9]. These include translation readthrough-inducing drugs (TRIDs), as well as nucleic acid-based approaches, including anticodon-edited suppressor tRNAs (ACE-tRNAs), mRNA editing, and DNA editing. To date, none of these approaches have received FDA approval for the treatment of any PTC disease. While several of the newer nucleic-acid based approaches appear to be promising, many problems must be overcome, in particular, off-target effects and effective delivery, before they can become approved treatments.

PTC suppression results from successful competition by an aminoacyl suppressor tRNA within a ternary complex (aa-tRNA.eEF1A.GTP, TC) with RFC for binding to a premature termination codon (PTC), leading to the elongation of the nascent peptide, a process known as readthrough (Figure 1). Recently, we developed a simple in vitro platform, denoted as PURE-LITE, for conducting detailed mechanistic studies of eukaryotic polypeptide elongation and termination [10], and used it to identify translation readthrough-inducing drugs (TRIDs) acting directly on the protein synthesis apparatus [11]. PURE-LITE takes advantage of the ability of the intergenic IRES of Cricket Paralysis Virus (CrPV-IRES) to form a tight complex with 80S ribosomes which is capable of initiating the cell-free synthesis of complete proteins in the absence of initiation factors. This allows for elongation and termination to be studied with the addition of just four factors, eEF1A and eEF2 for elongation and eRF1 and eRF3 for termination [12,13].

At present, the best characterized TRIDs are ataluren (Translarna), a hydrophobic substituted oxadiazole [14,15], and the highly polar class of aminoglycoside antibiotics (AGs) [2,16]. Ataluren is the only TRID which has been approved for treating patients suffering from a PTC disease, Duchenne muscular dystrophy (DMD). This approval has come from the European Medicines Agency. Ataluren is also approved for DMD treatment in Israel, Canada and Australia. We used the PURE-LITE platform to demonstrate that ataluren and G418, a potent AG, act orthogonally to stimulate readthrough; ataluren exclusively acts by competitively inhibiting RFC binding to the ribosome and G418 exclusively acts by stimulating the productive binding of near-cognate aminoacyl-tRNA to a stop codon (Figure 1). In addition, we newly identified two sites of ataluren binding within the ribosome, which, together, competitively inhibit RFC binding [13].

AGs such as G418 consistently induce significant improvement in functional readthrough [9] in cells isolated from patients suffering from a variety of PTC diseases. In spite of this, the clinical utility of AGs has, thus far, been restricted, in part, by their toxicities, which can arise from misreading of internal codons and the readthrough of normal stop codons, as well as other effects [2,3,17,18,19,20,21]. However, it is worth noting that an ongoing phase 2 clinical trial for a CF treatment with a molecularly engineered AG, ELX-02 (formerly known as NB124) has resulted in a significantly reduced toxicity [22]. In contrast, ataluren has been shown in several clinical trials to have very low toxicity, with little effect on the readthrough of normal stop codons [14], but it gives variable readthrough results in patient cells, with cells from some PTC diseases showing significant ataluren-induced readthrough, while others do not [23]. Additionally, recent efforts to inhibit termination activity by identifying small molecules, which induce either eRF1 [24] or eRF3 [25,26] depletion and potentiate the aminoglycoside induction of readthrough in cell-based assays, are of interest, although the toxicity of such molecules have not yet been fully explored.

Based both on our recent elucidation of ataluren’s mechanism of action as a competitive inhibitor of RFC and the identification of its binding sites on the ribosome, we think it may be possible to develop new TRIDs that would retain ataluren’s low toxicity, while displaying greater potency and generality in stimulating readthrough. Efforts in our laboratory are currently underway to achieve this using a combination of high-throughput screening (HTS) and a rational design. Below we present a detailed description of the plate reader assay we use both to screen small compound libraries for inhibition of RFC-dependent peptide release and to better understand the influence of codon identity and sequence context on RFC activity. The assay employs a eukaryotic pretermination ribosome complex containing a peptidyl-tRNA, FK*VRQ-tRNA^Gln^ in the P-site, denoted as Stop-POST5, where K* denotes a Lys residue labeled with Atto647 on the ε-amino group, neighboring an A-site containing a nonsense codon UGA (Figure 1). The RFC-dependent termination rate is determined by the time dependence of the change in Atto647 fluorescence anisotropy when the FK*VRQ peptide is released from the ribosome following peptidyl-tRNA hydrolysis. We first compare the peptide release rates obtained with the new assay to those of prior assays employing co-sedimentation and filter binding methods. We then show sample results demonstrating the utility of the HTS assay in determining the inhibition of termination by ataluren and some novel TRID inhibitors and the impact of varying stop codon identity and downstream sequence context on RFC activity.

## 2. Materials and Methods

### 2.1. Materials

The following materials were obtained or prepared as previously described [13]: (1) isoacceptor tRNAs charged with their cognate amino acids, *S. cerevisiae* tRNA^Phe^ and tRNA^Arg^; *E. coli* tRNA^Val^, tRNA^Lys^, and tRNA^Gln^; (2) *A. salina* 80S ribosomes and 40S and 60S subunits; (3) CrPV-IRES mRNAs, in which the N-terminal sequence encoding FKVRQ is followed by UGA CUA AUG (CrPV-IRES mRNA1), UGA GGA AAC (CrPV-IRES mRNA2), or UAG CGC CUG (CrPV-IRES mRNA3); (4) *S. cerevisiae* translation factors eEF1A and eEF2, which are used in the preparation of Stop-POST5 complexes, and eRF1, and eRF3, which are used in termination assays; (5) ataluren sodium salt.

#### 2.1.1. Other Materials

Full-length human eRF1 (heRF1) and human eRF3a (heRF3a) plasmids, which were obtained from PTC Therapeutics, were transformed into the BL21(DE3) Codon Plus (Agilent) strain in the presence of ampicillin. Single colonies were placed into 100 mL LB-amp media and grown overnight at 37 °C. The cultures were then diluted to 0.1 A_600_ and grown to 0.6 A_600_ in 2L LB-Amp media at 16 °C. 0.5 mM IPTG was added, and the cell cultures were incubated at 16 °C overnight. The cells were collected by centrifugation at 2700× *g* (4000 rpm in a GS3 rotor) for 20 min at 4 °C. A cell pellet (~5 g) was resuspended in heRF1/heRF3 Equilibration Buffer (100 mM HEPES-KOH, pH 7.4, 100 mM NaCl, 10% glycerol, 1 mM DTT, 50 mL) and lysed using a Qsonica sonicator at 30% using the largest sonicator tip for ten 15-second-pulse cycles, followed by 30 s cooling on ice. Cell debris was spun down at 27 k× *g* (15 k rpm in a SS34 rotor) for 15 min at 4 °C. Cell lysate (~ 55 mL) was loaded onto a 1.5 mL (3 mL slurry) TALON Superflow resin (Clontech), which had been equilibrated with heRF1/heRF3a Equilibration Buffer. The resin was then washed three times with 7.5 mL heRF1/heRF3a Wash Buffer (Equilibration Buffer plus 5 mM imidazole). Proteins were eluted in 0.5 to 1 mL fractions with heRF1/heRF3a Elution Buffer (Equilibration Buffer plus a 5–200 mM linear gradient of imidazole). Fractions with eRF1 or eRF3 were dialyzed against heRF1/heRF3 Equilibration Buffer overnight. Final protein concentrations were determined using A_280_. The extinction coefficients (ε_280_s) of eRF1 and eRF3 were calculated to be 33,600 cm^−1^ M^−1^ and 41,535 cm^−1^ M^−1^, respectively, using the Protparam Tool [27], and the protein sequences were obtained from the UniProt data base. The identities and purities of heRF1 and heRF3 were confirmed by SDS-PAGE. The candidate TRIDs (Figure 4D) were obtained from PTC Therapeutics. Guanidinomethyl-quinazoline.HCl (GMQ) was obtained from Sigma-Aldrich.

#### 2.1.2. Dye-Labeled [^3^H]-Lys-tRNA^Lys^ Formation

N-hydroxysuccinimide (NHS) esters of Atto-647, Cy3, and Alexa488, obtained from Sigma-Aldrich, were used to prepare dye-labeled [^3^H]-Lys-tRNA^Lys^. The reaction conditions for Atto-647 labeling were optimized with respect to the pH (10.8), time of incubation at 25 °C (4 min), DMSO (30%), and tRNA (0.1 mM), and NHS-ester (2 mM) concentrations to favor N^ε^ over N^α^ derivatization, as has been described in detail previously [13]. Identical conditions were used for labeling by the NHS esters of Cy3 and Alexa488. The stoichiometry of [^3^H]-Lys charging was typically 0.35 ± 0.05 Lys/tRNA. The stoichiometries of dye labeling per tRNA^Lys^ measured by UV absorption, were 0.40 ± 0.05 (Atto647, ε_647_: 120,000 mol^−1^ cm^−1^), 0.49 ± 0.02 (Cy3, ε_550_: 150,000 mol^−1^ cm^−1^) and 0.47 ± 0.07 (Alexa488, ε_494_: 73,000 mol^−1^ cm^−1^). Separation of the desired ε–labeled Lys-tRNA^Lys^ from the α–labeled and α,ε–di-labeled Lys-tRNA^Lys^ was not required since only the ε-labeled Lys-tRNA^Lys^ can be used for formation of the Stop-POST5 complex.

#### 2.1.3. Labeled Stop-POST5 Complexes

Atto647-labeled Stop-POST5 complexes were prepared from either purified 40S and 60S subunits or via high KCl treatment of 80S ribosomes as previously described [13]. Cy3- and Alexa488-labeled Stop-POST5 complexes were prepared exclusively from KCl-treated 80S ribosomes. 

### 2.2. Methods

#### 2.2.1. Peptide Release Assays

The co-sedimentation [12], plate reader [13] and Millipore filtration [13] assays measuring peptide release following the RFC-dependent hydrolysis of peptidyl-tRNA within the Stop-POST5 complex were performed as previously described.

#### 2.2.2. Thin Layer Electrophoresis

FK^*Atto647^VRQ peptide, which was produced by RFC-catalyzed hydrolysis of the Atto647-labeled Stop-POST5 complex, was isolated from the supernatants of co-sedimentation assays, spotted onto a TLC-cellulose plate (20 cm × 20 cm, Millipore sigma), dried, and electrophoresed at 300 V for 4 h using a mobile phase of pyridine (2.7 mL), acetic acid (110 mL), and water (550 mL) (pH 1.7), and covered by mineral oil [12]. Atto647 spots were identified using a Thypoon™ FLA 7000 reader (Appendix A).

#### 2.2.3. Mass Spectral Analysis of Released Labeled Peptide

The TLE-resolved FK^*Atto647^VRQ peptide was eluted with MeOH, and its mass was determined using a Waters Acquity LC/MS System with SQD Quadrupole MS (Appendix A).

## 3. Results

### 3.1. Incorporation of Fluorescent-Labeled Lys into the Stop-POST5 Complex

The stoichiometry of the peptidyl-tRNA/80S subunit, as measured by [^3^H]-Gln incorporation, varied between 0.3–0.4/ribosome for the ribosomes prepared from 40S and 60S subunits, and was about 40% lower for ribosomes prepared from high KCl-treated 80S ribosomes. These stoichiometries were minimally affected by the presence of the dye molecule on the Lys ε-amine (Table 1). The stoichiometry of Atto647 dye incorporated into Stop-POST5, which was measured by fluorescence intensity (Appendix A), was equal to 0.88 ± 0.03 of the value obtained for peptide stoichiometry by [^3^H]-Gln incorporation, which is likely due to minor amounts of non-dye labeled Lys-tRNA^Lys^ in the Atto647-Lys-tRNA^Lys^ preparation.

### 3.2. Dye Labeling Has No Substantial Effect on the Stoichiometry and Rates of RFC-Dependent Peptide Release from a Stop-POST5 Complex

As can be seen in Figure 2A, we used the co-sedimentation assay to compare the stoichiometry of the peptide release for the entire reaction (10′, 37 °C) for the labeled and unlabeled peptides. As measured in the supernatant, the release for the labeled peptide (75%) was similar, but somewhat lower, than the 90% level found for the unlabeled peptide. Rates of labeled and unlabeled peptide release, which were obtained under identical conditions, were determined by the co-sedimentation, TLE, plate reader, and filter binding assays. The apparent rate constant of Atto647-peptide release was determined by co-sedimentation and TLE analysis (Figure 2A), yielding similar values of 1.0–1.3 min^−1^. The MALDI mass spectral analysis confirmed the identity of the released fluorescent product as Atto647-FKVRQ, 1252 m/z (Appendix A). In contrast, the plate reader assay (Figure 2C) gave considerably lower, but virtually identical, rate constants for the three different dye-labeled pentapeptides: 0.27 ± 0.01 min^−1^ (Atto647); 0.26 ± 0.02 min^−1^ (Cy3); 0.24 ± 0.02 min^−1^ (Alexa488). Lastly, the filter binding assay measuring the unlabeled peptide release gave an apparent rate constant of 0.19 ± 0.05 min^−1^ (Figure 2D), which is similar to the values measured with the plate reader assays, showing that dye-labeling had only a minor effect on the rate of peptide release.

During termination, the comparatively rapid RFC-dependent peptidyl-tRNA hydrolysis process within the ribosome is followed by a much slower release of the free peptide from the ribosome [13]. We hypothesize that the difference between the faster rates determined in Figure 2B vs. those seen in Figure 2C,D are the result of an ultracentrifugation-induced release of free peptide from the ribosome.

### 3.3. Optimization of the Plate Reader Assay for High-Throughput (HT) Applications

#### 3.3.1. Results

Figure 2C,D show that the plate reader assay is a faithful reporter of the rate of peptide release from the ribosome, and can be used to study this reaction. In Figure 3, we present the results focused on optimizing standard procedures for high-throughput applications with respect to the pH, Stop-POST5 preparation, RFC preparation and concentration, and the addition of water miscible organic solvents. These results lead to the following conclusions:The pH optimum for the rate of peptide release falls in the range of 7.5–7.8 (Figure 3A,B);The apparent reactivity of Stop-POST5 complexes toward peptide release is somewhat higher for complexes prepared from purified 40S and 60S subunits than it is for complexes prepared by KCl treatment of 80S ribosomes, with the following EC_50_ and V_max_ values: 40S + 60S: EC_50_ 0.049 ± 0.003 µM, and V_max_ 0.49 ± 0.02 min^−1^; 80S: EC_50_ 0.029 ± 0.002 µM, and V_max_ 0.31 ± 0.02 (Figure 3C);Although eRF1 alone can catalyze peptide release, much higher rates were obtained when it was combined with eRF3 (Figure 3D);At an eRF1 concentration of 0.03 µM, the addition of 0.05 µM eRF3 essentially enables the formation of an active RFC complex (Figure 3E);Yeast and human RFCs have equal reactivity when they are freshly prepared. However, yeast RFC is not cold stable, whereas human RFC is (Figure 3F);Several water miscible solvents (DMSO, EtOH, DMF, and ACN) added up to 2% (*v*/*v*) in the reaction mixture had little or no effect on the RFC reactivity. This range can be extended to 5% for DMF and ACN (Figure 3G–J).

#### 3.3.2. Current Standard Conditions

Based on the results listed above, we are currently conducting our standard high-throughput assay at pH 7.5 and using a human RFC and a Stop-POST5 complex prepared by the KCl treatment of 80S ribosomes. The choice to use the latter method is dictated by the much greater ease of preparing large amounts of Stop-POST5 complex from 80S ribosomes as compared with preparations made from purified 40S and 60S subunits and the sufficiently high reactivity of the 80S preparation for acceptable use in the plate reader assay. In the early experiments, we demonstrated that use of a relatively high constant eRF3 concentration of 0.8 µM led to >90% formation of RFC from 0.01 to 0.4 µM eRF1. Going forward, the results in Figure 3E show that, at the lower eRF1 concentrations (≤0.06 µM) which we typically used in the plate reader assay, an eRF3 concentration of 0.1–0.2 µM is adequate. Finally, all of the results described in Figure 3 were performed at 25 °C to reduce the amount of solution evaporation.

### 3.4. Applications of the High-Throughput Assay

#### 3.4.1. Screening New TRID Candidates

Ataluren induces the readthrough of PTCs exclusively via the competitive inhibition of peptide release by RFC, and it does so by binding to functional sites on the ribosome, including the decoding center of the small subunit and the peptidyl transferase center of the large subunit [12,13]. However, such binding proceeds with low affinity, with EC_50_ values in the range of 200–400 µM, as measured either by the stimulation of readthrough or by inhibition of termination [12,13], and it is not unlikely that this low affinity accounts for the failure of ataluren to significantly improve the outcomes in clinical trials conducted on several PTC diseases. As we have noted earlier [12], while these EC_50_ values are 10–30 times higher than the ataluren concentrations employed in the growth media which are typically used in studies of ataluren stimulation of readthrough in intact cells or tissue cultures [15,28], this difference could be more apparent than real: i.e., ataluren, similar to other hydrophobic molecules, might be preferentially taken up by cells, resulting in a cellular concentration exceeding that which is present in the cell culture [29].

Our results suggest that it may be possible to obtain TRIDs which are more effective than ataluren in treating patients harboring PTC mutations by discovering compounds which bind to these functional ribosome sites with a much higher affinity, while preserving ataluren’s low toxicity. Toward this end we have begun the HT screening of compound libraries. Results with ataluren demonstrate a straightforward application of this screen, with time-dependent peptide release at several ataluren concentrations yielding traces which were well fitted with a single exponential decay function (Figure 4A). Plotting the observed rate constants against ataluren concentration showed a clear sigmoidal dependence (Figure 4B), which could be fit to a Hill equation, with a Hill *n* value of 3.0 ± 0.6 and a K_a_ value of 250 ± 20 µM, which is consistent with results of the single molecule experiments [13]. GMQ is a newly described TRID which shows the strong simulation of readthrough at <80 µM in Hela cell models harboring a nonsense mutation in p53 and at diverse sites on the dystrophin gene [30,31]. Our results with GMQ contrast markedly with those obtained for ataluren, showing no inhibition of the peptide release at concentrations of up to 200 µM (Figure 4C). The mechanism of GMQ stimulation of readthrough is yet to be determined, but it could arise from the stimulation of productive near-cognate tRNA binding, mimicking aminoglycosides, and/or inhibition of nonsense-mediated mRNA decay (NMD). More promising results were obtained in a collaborative project performed with PTC Therapeutics (Figure 4D). In this work, we determined the inhibitory activities toward peptide release of numbered compounds, which we tested at 300 µM. Some of the sample compounds with inhibitory activities comparable to or greater than that of 500 µM ataluren are shown.

We currently utilize ~0.3 pmol of Stop-POST5 programmed with CrPV-IRES mRNA1 per HT assay, and we can prepare the complex in 10 nmol quantities, making it feasible to carry out HT screens on commercial small molecule libraries containing thousands of molecules. We are exploring the development of simpler and less expensive methods of Stop-POST5 preparation at a scale of >100 nmol, which would facilitate the HTS of much larger libraries.

#### 3.4.2. Determining the Effect of Stop Codon Identity and Downstream Codon Sequence on Peptide Release Activity

In eukaryotic cell-based assays, PTC readthrough arises principally from functional near-cognate tRNA binding, in competition with RFC-catalyzed termination and NMD. As described in recent publications [32,33,34,35,36], the results of these assays support the major conclusions that readthrough is most favored by the UGA stop codon and that the sequence downstream from the stop codon has a greater influence on the probability of readthrough than the upstream sequence. We are employing our HT plate reader assay as part of an effort to test the predictions of the cell-based assays and to determine which disease-producing PTC mutations can best be treated by the TRIDs inhibiting RFC-dependent peptide release. We chose the sequence shown in Figure 4E, CrPV-IRES mRNA1, as a test sequence for most of the work presented in this paper. This sequence affords relatively high readthrough activity, thereby facilitating the performance of mechanistic studies [11,12,13], through its inclusion of both a UGA stop codon and a downstream CUA triplet which each favor readthrough [33]. In the sample results shown in Figure 4E–G, we compared the termination activities of ribosomes programmed with CrPV-IRES mRNA1 (Figure 4E) with those of ribosomes programmed with CrPV-IRES mRNA2 (Figure 4F) and CrPV-IRES mRNA3 (Figure 4G). The latter two mRNAs correspond to PTC mutant sequences in fibrillin 1 and CFTR protein, leading to Marfans syndrome [37] and cystic fibrosis [38], respectively. These two mRNAs have much a stronger interaction with RFC (K_m_^RFC^ values ≤ 3 nM) than they do with CrPV-IRES mRNA1 (K_m_^RFC^ ~ 50 nM), suggesting they would be more resistant to the TRID inhibitors of RFC activity. More comprehensive studies, including the examination of many additional stop codon-downstream sequence combinations and of the correlation between peptide release and readthrough activities, will be presented elsewhere.

## 4. Discussion

Here, we present a detailed account of our recent, fully reconstituted HT in vitro assay of termination activity. The reconstituted feature of the PURE-LITE platform permits the facile manipulation of experimental conditions, making it especially valuable for detailed studies of the termination mechanism, as demonstrated in our past publications [12,13] and in ongoing studies (Figure 4E–G). Its unique utility to achieve the high-throughput screening of potential small molecule inhibitors of termination lies in its ability to be quite precise about the mode of action of such inhibitors. An obvious limitation of the PURE-LITE platform is that it does not accurately recapitulate the environment present in living cells. As a consequence, the molecules identified in our HTS studies will require further testing prior to studies being conducted on animals. Recently described assays of termination activity which release full-length luciferases, measured in rabbit reticulocyte lysates [39,40], could serve as a useful step in this direction, along with well-established fluorescent reporter assays measuring the readthrough activity in intact cells [28,41,42], including those derived from patients suffering from a variety of PTC diseases.

The sample results presented in Figure 4E–G provide a clear indication of the dependence of the strength of RFC interaction with a Stop-POST5 complex on the identity of the stop codon and the adjacent downstream sequence. These results suggest that medically effective readthrough of PTCs which interact strongly with RFCs may be best achieved by using TRIDs, which interfere with RFC binding, to potentiate the effectiveness of other nonsense suppressors that directly stimulate productive aa-Suppressor tRNA binding (Figure 1). This approach has been demonstrated by ourselves [12] and others [24,25,26] in potentiating G418-induced readthrough. It might also be useful in potentiating readthrough induced by ACE- tRNAs [42,43] as well, by reducing the competition by RFC for binding to a PTC site.

## Figures and Tables

**Figure 1 biomolecules-13-00242-f001:**
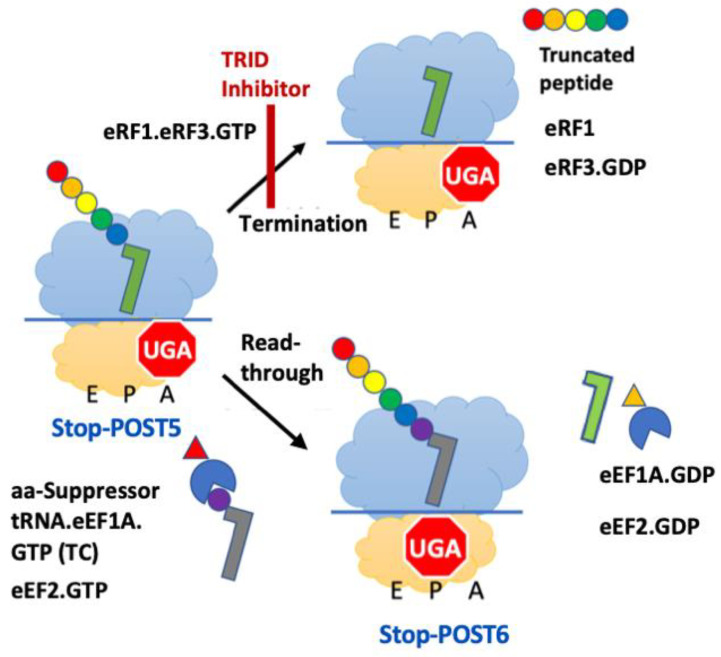
Competition for binding to a UGA PTC between release factor complex, leading to peptidyl-tRNA hydrolysis and nascent peptide release (i.e., termination), and aminoacyl suppressor tRNA ternary complex, leading to readthrough. In our standard HT assay, Stop-POST5 contains FK*VRQ-tRNA in the P-site, where K* denotes a Lys residue labeled with Atto647 on the ε-amino group.

**Figure 2 biomolecules-13-00242-f002:**
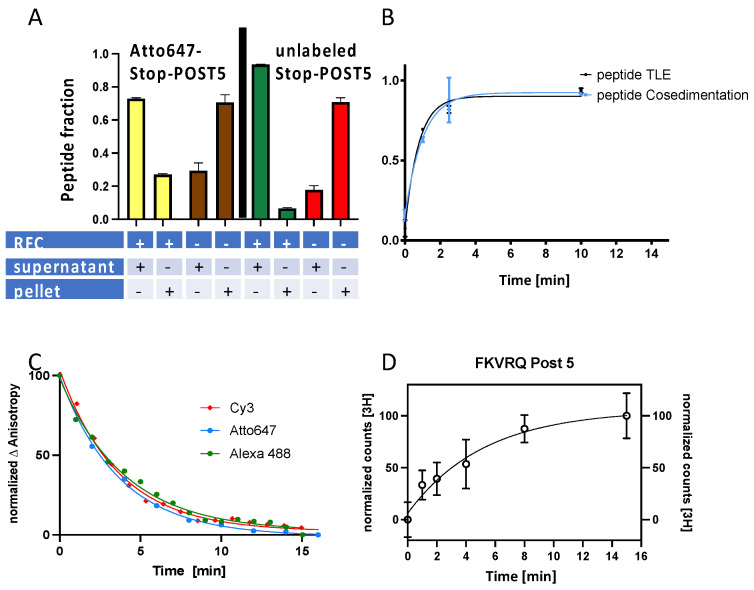
Measures of pentapeptide release at pH 7.5. In all of the experiments, Stop-POST5, 0.05 µM, was prepared using KCl-treated 80S ribosomes programmed with CrPV-IRES mRNA1, and GTP was present at 1 mM. (**A**) Release of Atto647-labeled and unlabeled pentapeptides measured by co-sedimentation. Conditions: 10 min incubation, 37 °C, eRF1 0.2 µM, and eRF3 0.4 µM. (**B**) Rates of Atto647-labeled pentapeptide release measured by co-sedimentation and TLE assays. Conditions: eRF1 0.1 µM, eRF3 0.2 µM, and 25 °C. (**C**) Rates of dye-labeled pentapeptide release as measured in the plate reader assay using three different dyes. Apparent rate constants were determined by fitting traces to a one phase decay model (Graphpad prism). Conditions: eRF1 0.2 µM, eRF3 0.4 µM, and 25 °C. (**D**) Rate of unlabeled pentapeptide release measured by Millipore filtration, calculated using both filtrate and filter measurements. Conditions: eRF1, 0.2 μM, eRF3 0.8 μM, and 25 °C. Error bars: A and B, ± average deviations, n= 2; D, ± standard deviations, n = 8 for all points except for the 4 min measurements, for which n = 11.

**Figure 3 biomolecules-13-00242-f003:**
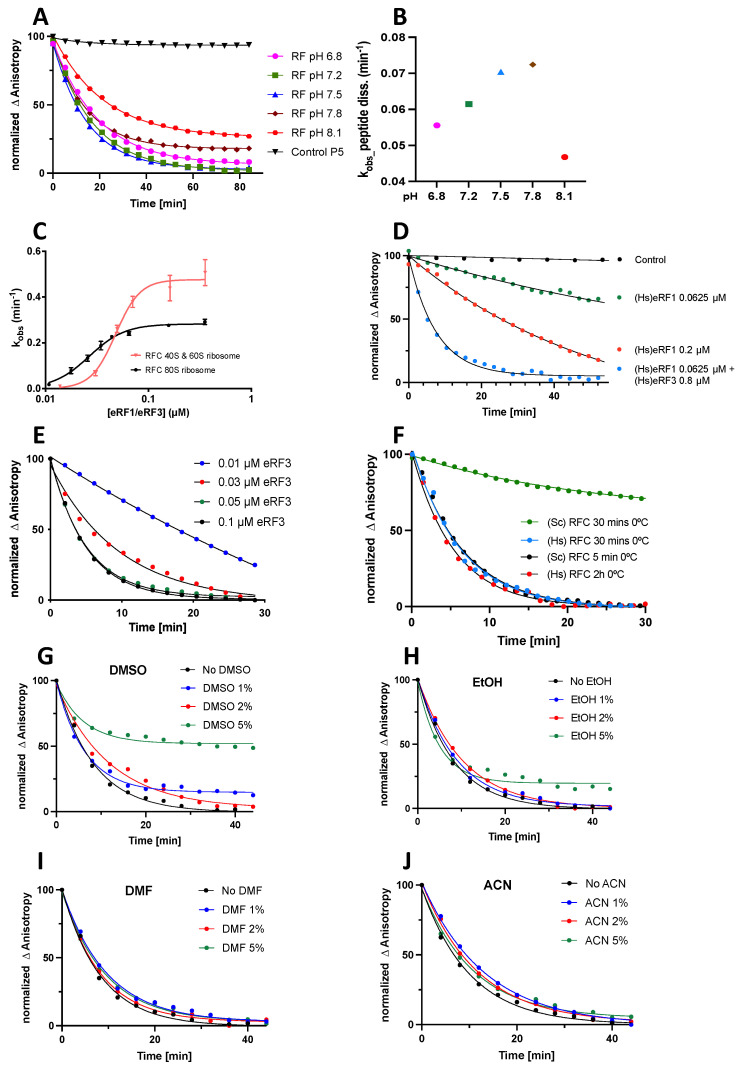
Optimization of the plate reader termination assay. All determinations were made at 25 °C. Except as otherwise indicated below, Stop-POST5 (0.05 µM) was prepared from KCl-treated 80S ribosomes programmed with CrPV-IRES mRNA1, pH was 7.5 and the GTP was 1 mM. (**A**) pH-dependent traces. Conditions: Stop-POST5, 0.05 µM, eRF1 0.05 µM, eRF3 0.2 µM, and n = 2. The traces are normalized to pH 7.5, making the smaller anisotropy changes observed at a higher pH more apparent. (**B**) pH-dependent apparent rate constants in **A**. (**C**) Comparison of rates for Stop-POST5 prepared from either 40S + 60S subunits or KCl treated 80S. Conditions: eRF1 0.025–0.40 µM and eRF3 0.8 µM. Error bars are average deviations, n = 2. (**D**) eRF3 stimulation of eRF1 termination activity, eRF1 and eRF3 concentrations as shown, n = 2. (**E**) Dependence of termination rate on eRF3 concentration. Conditions: Stop-POST5 0.03 µM, eRF1 0.05 µM, eRF3 0.01–0.1, and n = 2. (**F**) Comparison of the cold stabilities of yeast and human RFCs. Conditions: eRF1 0.05 µM, eRF3 0.8 µM, and n = 2. (**G**–**J**) Effects on termination activity of adding small percentages of the specified water-miscible organic solvents to the reaction medium. Conditions: eRF1 0.05 µM, eRF3 0.1, and n = 2. In each figure, traces are normalized to the trace containing no added solvent, making clear the smaller anisotropy change observed with either DMSO or EtOH.

**Figure 4 biomolecules-13-00242-f004:**
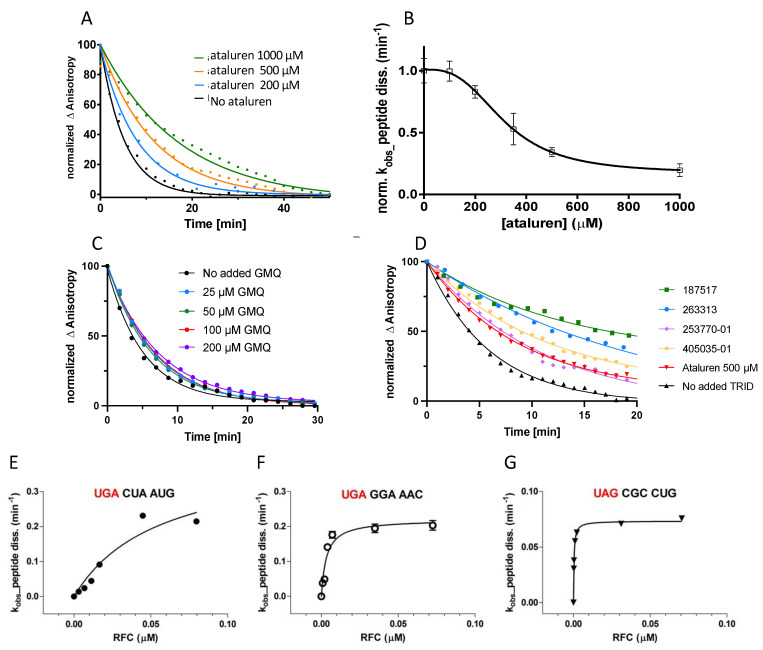
Applications of the high-throughput peptide release assay. All determinations were performed at 25 °C, pH 7.5, and 1 mM GTP. Stop-POST5 was prepared from KCl-treated 80S ribosomes programmed with either CrPV-IRES mRNA1 (**A**–**E**), CrPV-IRES mRNA2 (**F**), or CrPV-IRES mRNA3 (**G**). Stop-POST5 was present at either 0.05 µM (**A**–**D**) or 0.04 µM (**E**–**G**). (**A**) Sample ataluren inhibition traces. (**B**) Inhibition as a function of ataluren concentration. Conditions: eRF1 0.0625 µM, eRF3 0.8 µM, and n = 2. (**C**) Failure of GMQ to inhibit peptide release, 25–200 µM. Conditions: eRF1 0.0625 µM, eRF3 0.8 µM, and n = 2. (**D**) Inhibition by candidate TRIDs added at 300 µM compared with ataluren added at 500 µM. Conditions: eRF1 0.0625 µM, eRF3 0.8 µM, n = 2. (**E**–**G**) Sample results showing strong influence of stop codon (shown in red) and downstream codons on EC_50_ and V_max_ values for RFC catalysis of peptide release.

**Table 1 biomolecules-13-00242-t001:** FK*VRQ-tRNA^Gln^ stoichiometry per Stop-POST5 complex.

80S Ribosome Preparation ^a^	Fluorescent Label	Relative Peptide/80S ^b^
40S + 60S	None	1.00 ^c^
	Atto647	0.90 ± 0.10
KCl-treated 80S	None	0.56 ± 0.07
	Atto647	0.56 ± 0.12
	Alexa 488	0.53 ± 0.03
	Cy3	0.75

^a^ See Materials and Methods; ^b^ Measured by [3H]-Gln incorporation; ^c^ Stoichiometry varied from 0.3–0.4/ribosome.

## Data Availability

All relevant data for this paper are presented in the main text or in Appendix A. Any questions should be sent to B.S.C.

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
