# Peer review of "A High-Throughput Assay for In Vitro Determination of Release Factor-Dependent Peptide Release from a Pretermination Complex by Fluorescence Anisotropy—Application to Nonsense Suppressor Screening and Mechanistic Studies"

_biomolecules, 2023, doi:10.3390/biom13020242_

Round 1

Reviewer 1 Report

As expected from a lab that has consistently published high quality work, this ms reports useful and significant results on an important topic. The same lab is involved in a comprehensive investigation of readthrough enhancing drugs, and the present ms fits well into their approach. The ms is clearly written, and somewhat unusually I have no criticisms.

Author Response

None needed

Reviewer 2 Report

In this manuscript authors describe an HT method for screening compounds inducing stop codon readthrough. The method is well described and the results are convincing.

Minor issues:

introduction page 2 line 43 "and inhalation" it is not clear, the sentence should be amended.

In the introduction section the effect of TRIDs on general translation on on the failed correct recognition of stop codons should be discussed.

In the discussion section a more extensive discussion of the differences of the described HT assay compared to cellular (human) translational machinery should be included also taking into account the very high concentrations required for known drugs such as ataluren in the assay.

Author Response

Minor issues:

introduction page 2 line 43 "and inhalation" it is not clear, the sentence should be amended.

Response: We agree and have deleted “and inhalation”

In the introduction section the effect of TRIDs on general translation on on the failed correct recognition of stop codons should be discussed.

Response: We have added new text (lines 84, 88, 89) and new references (new #s 14, 15, 21) to clearly emphasize that AGs promote readthrough at normal stop codons whereas ataluren does not. These new references, along with others previously cited, will direct the interested reader to more detailed considerations of TRID effects on normal stop codon readthrough.

In the discussion section a more extensive discussion of the differences of the described HT assay compared to cellular (human) translational machinery should be included also taking into account the very high concentrations required for known drugs such as ataluren in the assay.

Response: We have addressed this point with new text added to Section 3.4.1 (lines 309-314).

Reviewer 3 Report

Major comments:

1.       According to the study, the identity of the stop codon and the adjacent downstream sequence play an important role in readthrough. However, the manuscript only compares the effects of three stop codon-downstream sequence combinations on peptide release and readthrough activities. It would be better if the authors could provide more examination about additional stop codon-downstream sequence combinations.

2.       The PURE-LITE platform used in this study is only an in vitro test platform, which cannot accurately recapitulate the environment present in living cells. Therefore, it is suggested that the author could further evaluate this new high throughput screening technique in living cells or cells isolated from patients suffering from a variety of PTC diseases.

Minor comments:

1.     In Page 7, Section 3.3.2, the authors say that “the reasonably high reactivity of the 80S preparation in the plate reader assay”. But in Figure 3C, the rates for Stop-POST5 prepared from 40S + 60S subunits is much higher than KCl treated 80S. It is recommended to explain the definition of “reasonably high”.

2.     In Page 7, only the concentrations of eRF1 and eRF3 are given in the reaction system. However, according to Page 2 Line 65 and Page3 Line 118, there are 4 factors involve. Therefore, it would be helpful if the authors could supplement the concentrations of elongation factors Eef1A and eEF2.

3.     It is mentioned on page 7 that the EC50 values of Ataluren is in the range of 200-400μM, and its affinity for binding ribosome functional sites is low. Then what is the appropriate EC50 value? It is suggested that the authors could further explain the relationship between the efficacy of TRIPs and indicators such as EC50 and Vmax values.

4.     It is recommended to explain the last sentence of the manuscript “It should also be useful in potentiating readthrough induced by ACE- tRNAs as well”, since ACE-tRNAs and TRIDs involved in this study are two anti-PTC methods with different mechanisms.

Author Response

Major comments:

  1. According to the study, the identity of the stop codon and the adjacent downstream sequence play an important role in readthrough. However, the manuscript only compares the effects of three stop codon-downstream sequence combinations on peptide release and readthrough activities. It would be better if the authors could provide more examination about additional stop codon-downstream sequence combinations.

Response: We agree that results for more examples of stop codon-downstream sequence combinations will be required to fully elucidate their importance in determining readthrough. However, such results are not within the scope of this MS, which rather is to provide a detailed description of the plate-reader assay and some examples of its application (lines 100-103). Our work currently underway to provide more comprehensive results are referred to in lines 406-408.

  1. The PURE-LITE platform used in this study is only an in vitro test platform, which cannot accurately recapitulate the environment present in living cells. Therefore, it is suggested that the author could further evaluate this new high throughput screening technique in living cells or cells isolated from patients suffering from a variety of PTC diseases.

Response: We agree with the reviewer’s comment. We have added to our earlier discussion of this topic a direct reference to cells derived from patients suffering from PTC diseases (lines 458-459).

Minor comments:

  1. In Page 7, Section 3.3.2, the authors say that “the reasonably high reactivity of the 80S preparation in the plate reader assay”. But in Figure 3C, the rates for Stop-POST5 prepared from 40S + 60S subunits is much higher than KCl treated 80S. It is recommended to explain the definition of “reasonably high”.

Response: We agree and changed “reasonably high reactivity of the 80S preparation in the plate reader assay” to “sufficiently high reactivity of the 80S preparation for acceptable use in the plate reader assay”. (lines 293-294)

  1. In Page 7, only the concentrations of eRF1 and eRF3 are given in the reaction system. However, according to Page 2 Line 65 and Page3 Line 118, there are 4 factors involve. Therefore, it would be helpful if the authors could supplement the concentrations of elongation factors Eef1A and eEF2.

Response: As described on p. 2, line 65, only eRF1 and eRF3 are used in the termination assays. The roles of eEF1A and eEF2 in this work is simply to prepare Stop-POST5 complexes, using a previously described procedure (Ref. 13). We have added text to Section 2.1 to make this point explicit (lines 122, 123).

  1. It is mentioned on page 7 that the EC50 values of Ataluren is in the range of 200-400μM, and its affinity for binding ribosome functional sites is low. Then what is the appropriate EC50 value? It is suggested that the authors could further explain the relationship between the efficacy of TRIPs and indicators such as EC50 and Vmax values.

Response: We have added a phrase (lines 306, 307) explaining that EC50 values for ataluren are somewhat different depending on whether one is measuring effects on readthrough or termination. Each of the Vmax and EC50 values provided in the MS have their standard meanings.

  1. It is recommended to explain the last sentence of the manuscript “It should also be useful in potentiating readthrough induced by ACE- tRNAs as well”, since ACE-tRNAs and TRIDs involved in this study are two anti-PTC methods with different mechanisms.

Response: We have added the phrase “by reducing the competition by RFC for binding to a PTC site.” (lines 470, 471)